# ATAC-seq with unique molecular identifiers improves quantification and footprinting

Tao Zhu [1,3], Keyan Liao[1,3], Rongfang Zhou[1,3], Chunjiao Xia[1] & Weibo Xie [1,2]✉

ATAC-seq (Assay for Transposase-Accessible Chromatin with high-throughput sequencing) provides an efficient way to analyze nucleosome-free regions and has been applied widely to identify transcription factor footprints. Both applications rely on the accurate quantification of insertion events of the hyperactive transposase Tn5. However, due to the presence of the PCR amplification, it is impossible to accurately distinguish independently generated identical Tn5 insertion events from PCR duplicates using the standard ATAC-seq technique. Removing PCR duplicates based on mapping coordinates introduces increasing bias towards highly accessible chromatin regions. To overcome this limitation, we establish a UMI-ATAC-seq technique by incorporating unique molecular identifiers (UMIs) into standard ATAC-seq procedures. UMI-ATAC-seq can rescue about 20% of reads that are mistaken as PCR duplicates in standard ATAC-seq in our study. We demonstrate that UMI-ATAC-seq could more accurately quantify chromatin accessibility and significantly improve the sensitivity of identifying transcription factor footprints. An analytic pipeline is developed to facilitate the application of UMI-ATAC-seq, and it is available at https://github.com/tzhu-bio/UMI-ATAC-seq.

[1] National Key Laboratory of Crop Genetic Improvement, Huazhong Agricultural University, 430070 Wuhan, China. [2] Hubei Key Laboratory of Agricultural Bioinformatics, Huazhong Agricultural University, 430070 Wuhan, China. [3] These authors contributed equally: Tao Zhu, Keyan Liao, Rongfang Zhou. ✉email: weibo.xie@mail.hzau.edu.cn

n eukaryotes, nuclear DNA is tightly packed into nucleosomes. The interaction of transcription factors with DNA and the exercise of transcriptional regulatory functions require or eventually lead to the dissociation of nucleosomes and form nucleosome-depleted regions or called open chromatin regions[1]. Thus, open chromatin regions are considered to be the primary location of transcription factor binding sites[2,3]. ATAC-seq provides an efficient way to analyze nucleosome-depleted regions of the genome and has been applied widely to identify transcription factor footprints[4–6]. Both applications rely on the accurate quantification of insertion events of the hyperactive transposase Tn5. In the assay, the density of Tn5 insertion events reflects the chromatin accessibility of a region, while some small regions with sudden drops of insertion events in a highly accessible region might indicate footprints of bound transcription factors.

As with most sequencing methods, ATAC-seq uses PCR amplification to ensure sufficient DNA fragments for sequencing on Illumina platforms[7]. However, fragments are amplified with different efficiencies, resulting in an excessive number of specific fragments in the final library and bias for a number of analysis[8]. To eliminate this bias, we usually remove the PCR duplicates by identical genome coordinates before the advanced analysis. However, this deduplication mode is flawed in some situations. For example, as the sequencing depth increases, the probability of the same fragment being generated from different Tn5 insertion events will increase. For species with a small genome size or many tandem repeats, this phenomenon is exacerbated because they have a much higher probability of producing the identical fragments before PCR amplification[7]. In this case, removing PCR duplicates with mapping coordinates will remove the usable fragments that are not generated by PCR amplification, resulting in information loss, and introducing another kind of bias.

UMIs, typically random oligonucleotides ligated to each molecule, are widely used to accurately detect the PCR duplicates and quantitative transcripts[9–12]. The identical fragments with the same UMI are thought to be generated by PCR amplification. And the identical fragments with different UMIs, on the other hand, are assumed to be derived from different cells or different genome coordinates. UMI is demonstrated to be effective in counting the absolute number of individual molecules, and has been applied to various sequencing methods[9,13,14].

Here, we report a UMI-ATAC-seq technique by incorporating UMIs with standard TruSeq sequencing adapters into standard ATAC-seq procedures. The UMI-ATAC-seq library is compatible for sequencing on the Illumina platforms. Our results show that UMI-ATAC-seq could eliminate PCR deduplication bias especially in highly accessible chromatin regions. Finally, we demonstrate that UMI-ATAC-seq could more accurately quantify chromatin accessibility and significantly improve the sensitivity of identifying footprints.

## Results

### Increased PCR duplication rates in highly accessible chromatin regions indicate bias for quantification and footprint identification in conventional ATAC-seq.
We downloaded ATAC-seq data from Arabidopsis thaliana[5] and human[15], which are generated using standard ATAC-seq protocol. We divided the Arabidopsis and human genome into 250 bp bins separately and counted the total number of reads mapped in each bin and the number of reads marked as PCR duplicates by their identical mapping coordinates, respectively. When we evaluated PCR duplication rates in regions with different chromatin accessibility, we found that highly accessible chromatin regions usually include more PCR duplicates (Fig. 1a, b). Obviously, such trends could not be simply explained by PCR amplification efficiencies. Hence

the common practice, computational removal of PCR duplicates based only on their identical mapping coordinates, might introduce bias for quantification in ATAC-seq.

In addition, we identified and counted the putative footprints in each 250 bp genomic bin. The results show that most of the footprints are identified in highly accessible regions and thus might be more affected by the way of removing PCR duplicates (Fig. 1c, d).

### Adapting standard ATAC-seq procedures to incorporate UMIs.
To assess the bias of PCR deduplication, we adapted standard ATAC-seq procedures to incorporate UMIs by redesigning a set of sequencing adapters and named our improved method UMI-ATAC-seq (Fig. 2). Each of the new adapters consists of a 19-bp mosaic end (ME) sequence in the 3′ end for the binding of Tn5, a UMI sequence at least 6 bp in the middle, and a standard TruSeq sequencing adapter in the 5′ end for compatibility with standard sequencing to allow mixing with other kinds of samples. (Fig. 2c). UMI-ATAC-seq library is compatible for sequencing on the Illumina platforms, such as HiSeq and NovaSeq systems. Additional barcodes could be added to the 5′ end of Truseq sequence or 3′ of the UMI sequence for sample multiplexing.

In UMI-ATAC-seq, the first read (Read 1) will start from the UMI sequence. Illumina sequencing systems utilize the first 25 cycles of Read 1 for cluster identification and quality metrics calculations[16]. To accommodate this requirement, when solo libraries of UMI-ATAC-seq were loaded onto the flow cell, we could mixed a set of adapters with UMI sequence of four lengths (e.g., 6 bp, 13 bp, 20 bp, and 25 bp in our study) to insure enough base diversity in the first 25 cycles (Fig. 2a).

The procedures of assembling the Tn5 transposon complex could refer to ref. [17] and all other steps are the same as for conventional ATAC-seq[18], except that the PCR primers need to be replaced when constructing the libraries (Fig. 2b). Overall, the UMI-ATAC-seq technique can be readily mastered by experienced researchers.

### The feasibility of UMI-ATAC-seq.
We obtained high quality data from seven rice young panicle samples using UMI-ATAC-seq. An analytic pipeline was developed to facilitate the application of UMI-ATAC-seq. Briefly, UMIs were extracted and appended to the read names in FASTQ files. We then located the ME sequences and trimmed the reads to the end of the ME sequences. After mapping the reads to the genome, a script was used to remove duplicate fragments with both identical mapping coordinates and UMI sequences.

We observed similar insert size distributions of sequenced fragments as the standard ATAC-seq, reflecting that Tn5 integration events are enriched in regions between nucleosomes (Supplementary Fig. 1a). The Tn5 integration sites are highly enriched around the Transcription Start Site (TSS) and the TSS enrichment scores of all samples are greater than thirteen (Supplementary Fig. 1b and Supplementary Table 2). And the fraction of reads in called peak regions (FRiP) are more than 60% (Supplementary Table 2). Taken together, these data demonstrated that UMI-ATAC-seq is feasible to generate high quality chromatin accessibility data.

### The characteristics of rescued reads.
When incorporating UMIs into ATAC-seq, we could distinguish and only remove the duplicated reads generated by PCR amplification and keep reads from coincidental Tn5 insertion events by their different UMI sequences. For the rice young panicle samples analyzed using UMI-ATAC-seq, the results show that UMI-ATAC-seq

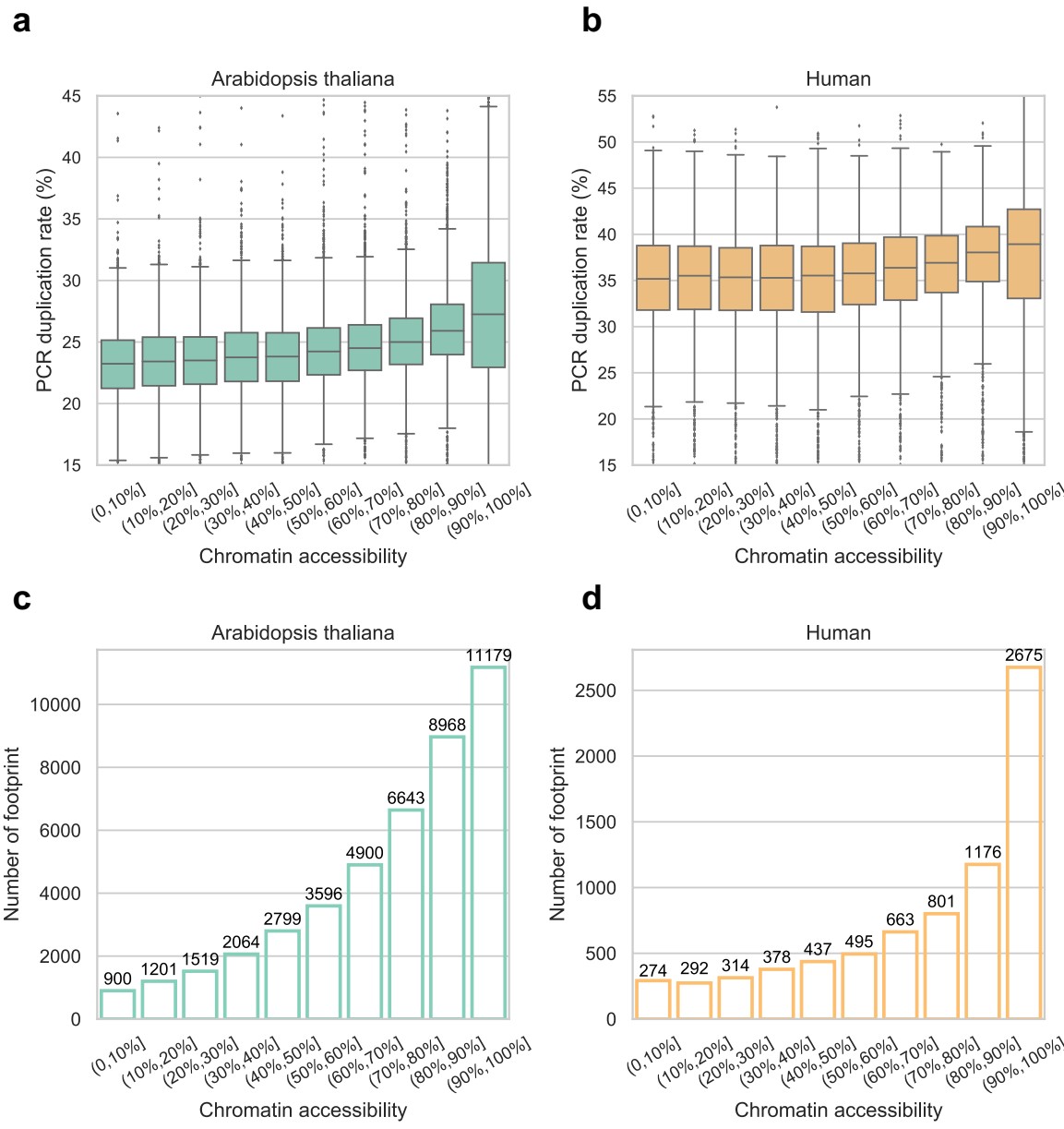

**Fig. 1 PCR duplication rates and the number of identified footprints in regions with different chromatin accessibility. a** Box plot illustrating the distribution of PCR duplication rates in regions with different chromatin accessibility in Arabidopsis seedlings. We divided the top 10.0% of the most accessible 250 bp bins into deciles for analysis. **b** Box plot illustrating the distribution of PCR duplication rates in regions with different chromatin accessibility in human HEK293 cells. We divided the top 1.6% (Tn5 insertions > 350) of the most accessible 250 bp bins into deciles for analysis. **c** The number of identified footprints in bins with different chromatin accessibility in Arabidopsis. ATAC-seq data and the grouping of bins are the same as in **a**. **d** The number of identified footprints in bins with different chromatin accessibility in human. ATAC-seq data and the grouping of bins are the same as in **b**.

could rescue about 20% of reads that would be mistaken as PCR duplicates in standard ATAC-seq (or ~6% of the total mapped reads) (Supplementary Table 2). We divided the genome into 250 bp bins and counted the rescued reads in each bin. We observed that the more accessible the region, the more reads UMI-ATAC-seq could rescue (up to 30% or more in some highly accessible regions, Supplementary Fig. 2a). Of the rescued reads in sample C019, 81.8% come from the top 2.0% of the most accessible bins in the rice genome. Moreover, the rescued reads are enriched in regions susceptible to copy number variations or genome assembly errors, such as rDNA regions, transposons, retrotransposons, and telomeres (Supplementary Fig. 2b, c).

**UMI-ATAC-seq improves quantification of chromatin accessibility**. To assess the impact of incorporating UMIs on quantification, we quantified the Tn5 insertions of each 250 bp bin by mapping coordinate-based deduplication (CD), UMI-based deduplication (UD), and no deduplication (ND), respectively, and compared them with each other. The results show that the correlation between UD and ND is slightly higher than UD and CD ($R^2 = 0.995$ and 0.949, respectively) (Fig. 3b, c); we obtained similar results with the other six samples (Supplementary Fig. 3b, c). As the accessibility of the region increases, the quantification by CD is more deflected from UD (Fig. 3c and Supplementary Fig. 3c). We further examined the UMI duplication rate in regions with different chromatin accessibility. We found that the

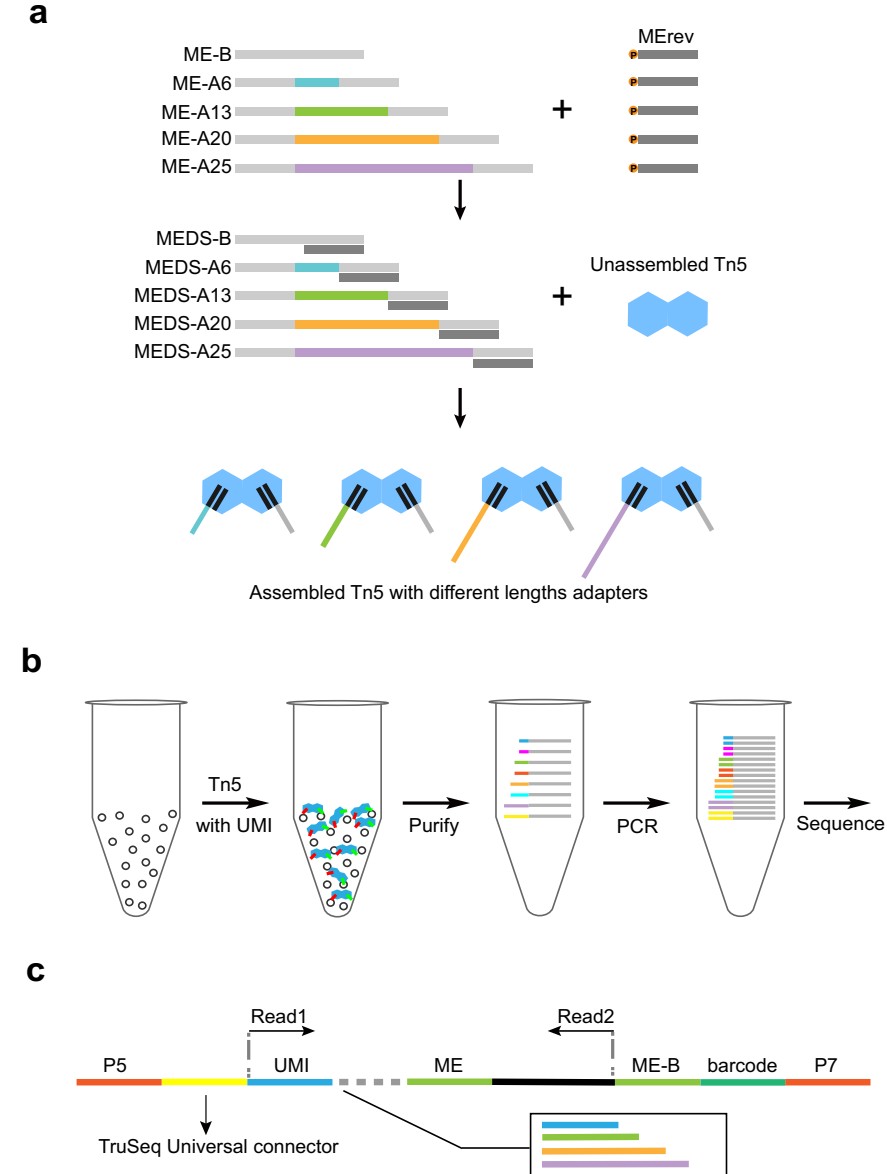

**Fig. 2 Incorporating UMIs to standard ATAC-seq procedures. a** The diagram of assembling Tn5 transposases with mixed adapters of four lengths. **b** The workflow of UMI-ATAC-seq. **c** The library structure of UMI-ATAC-seq. See Methods for details.

UMI duplication rate is stable, unlike the coordinate-based duplication rate, which is increased in highly accessible chromatin regions (Fig. 3d). These results suggest that UMI-based deduplication could improve quantification in ATAC-seq.

**UMI-ATAC-seq has a slight effect on peak calling.** Identifying accessible chromatin regions (or peak calling) is a major step of ATAC-seq data analysis. To evaluate the impact of incorporating UMIs on peak calling, we called peaks using MACS2[19] with different parameters reported by different studies[5,20,21]. We compared the number of peaks, $P$-values, and fold-enrichments obtained with datasets processed in CD and UD modes, respectively. And we found only small differences in the number of peaks between the CD and UD datasets for various peak calling parameters (Supplementary Fig. 4).

However, for most of the highly significant peaks, the results of the UD dataset are slightly better than the CD dataset in terms of the $P$-value and the fold-enrichment (Fig. 4a, b). Furthermore, we found that wider peaks and peaks with higher fold-enrichment

had more presumed PCR duplicates in CD mode, and we could eliminate such bias in UD mode (Fig. 4c, d). These results suggest that while UMI-ATAC-seq has only a slight effect on peak calling, it might improve the estimation of fold-enrichment of peaks, as it can eliminate PCR deduplication bias.

**UMI-ATAC-seq improves the sensitivity of footprint identification.** Footprint identification is a common analysis to decipher the transcription factor binding sites in ATAC-seq assays[22]. We identified footprints by pyDNase[23] using the rice UMI-ATAC-seq datasets processed in CD and UD modes separately. We observed that the number of footprints identified in UD datasets was significantly greater than CD datasets (Fig. 5a and Supplementary Fig. 5a). Although only ~6% of the total reads were rescued, they contributed to the identification of an additional 50% or more of footprints. Comparing the scores of overlapping footprints, we found that the scores of footprints in UD datasets were almost always higher than CD datasets. (Fig. 5b and Supplementary Fig. 5b).

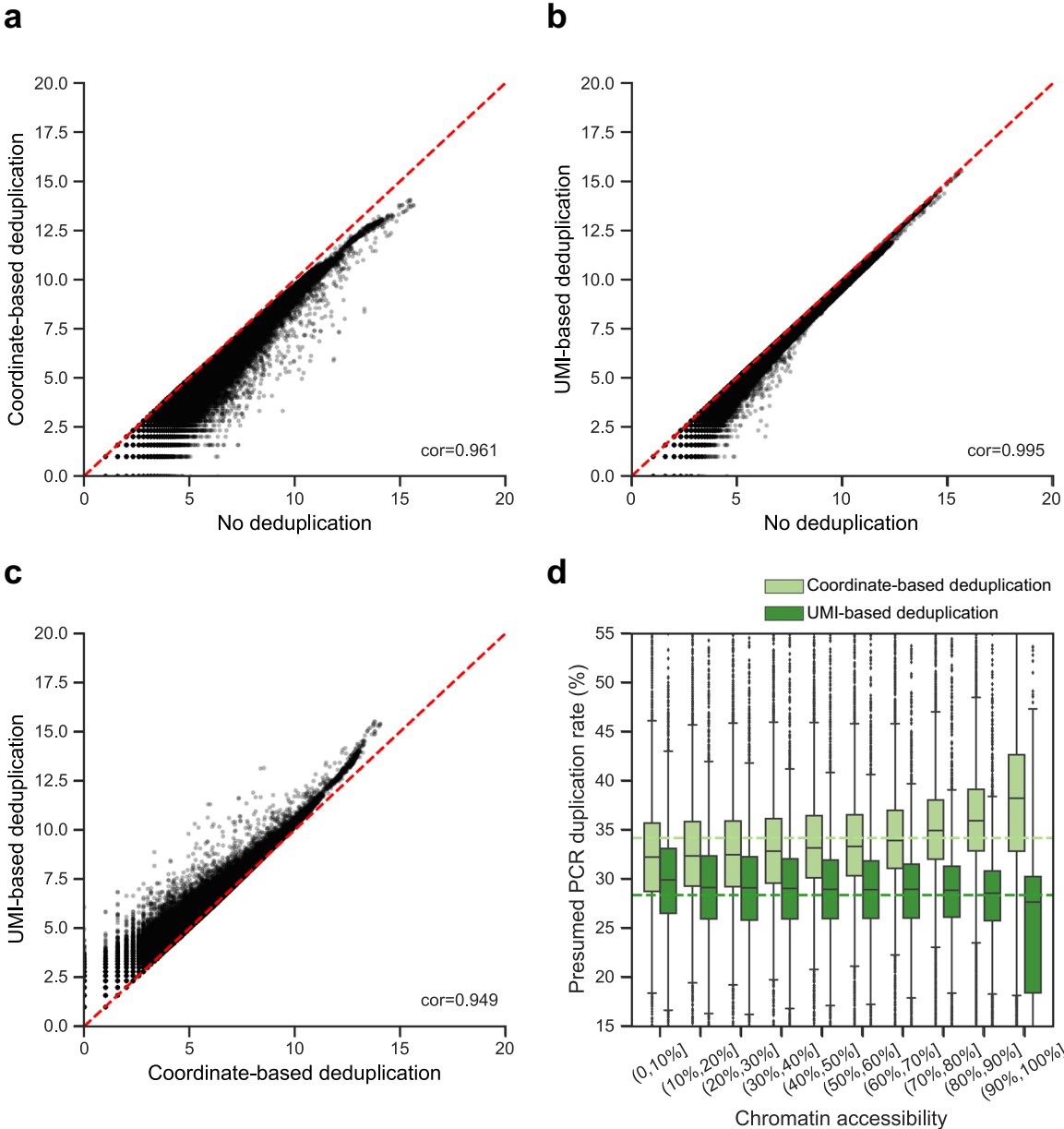

**Fig. 3 The impact of deduplication in different ways on quantification.** All analyses were performed using UMI-ATAC-seq data from rice sample C019. **a–c** Scatter plots contrasting quantification results of no deduplication (ND), coordinate-based deduplication (CD), and UMI-based deduplication (UD) datasets. Each point is the number of Tn5 insertions in a 250 bp bin (log base 2). **d** Box plot illustrating the distribution of the presumed PCR duplication rates in regions with different chromatin accessibility. We divided the top 4.4% (Tn5 insertions > 100) of the most accessible 250 bp bins into deciles for analysis. The dashed lines represent the averages of the corresponding presumed PCR duplication rates.

We checked the frequency of Tn5 insertions around the identified footprints using IGV (Integrative Genomics Viewer)[24], and confirmed that the patterns of footprints were more apparent with UD datasets (Fig. 5c). For UD-only footprints, many of the independent Tn5 insertions were removed as PCR duplicates in CD datasets, making their footprint scores below the threshold and therefore unidentifiable in CD datasets. While for CD-only footprints, the corresponding footprint scores in UD datasets were actually higher than in CD datasets. The footprints could not be identified in UD datasets merely because they are likely omitted by the algorithm of pyDNase, which identified other close footprints around and therefore prevented the identification of the relevant footprints (e.g., region VI in Fig. 5c).

Moreover, we calculated the footprint depth (FPD) values for ND, CD, and UD datasets respectively and compared the differences of FPD between each other. A bigger FPD value could be interpreted as decreased Tn5 insertions in the footprint region compared to its flanking regions.(Fig. 5d) The results show that FPD values in UD datasets are generally higher than CD datasets, indicating that patterns of footprints are more apparent in UD mode (Fig. 5e and Supplementary Fig. 6a). In addition, we scanned all potential transcription factor binding motifs across the genome using FIMO with known position weight matrices ($P$-value $< 10^{-4}$)[25]. We averaged the Tn5 insertions across all predicted binding sites of transcription factor AP2/ERF and TCP1 that overlapped with the common pyDNase footprints identified in both CD and UD modes and located in the top 25%

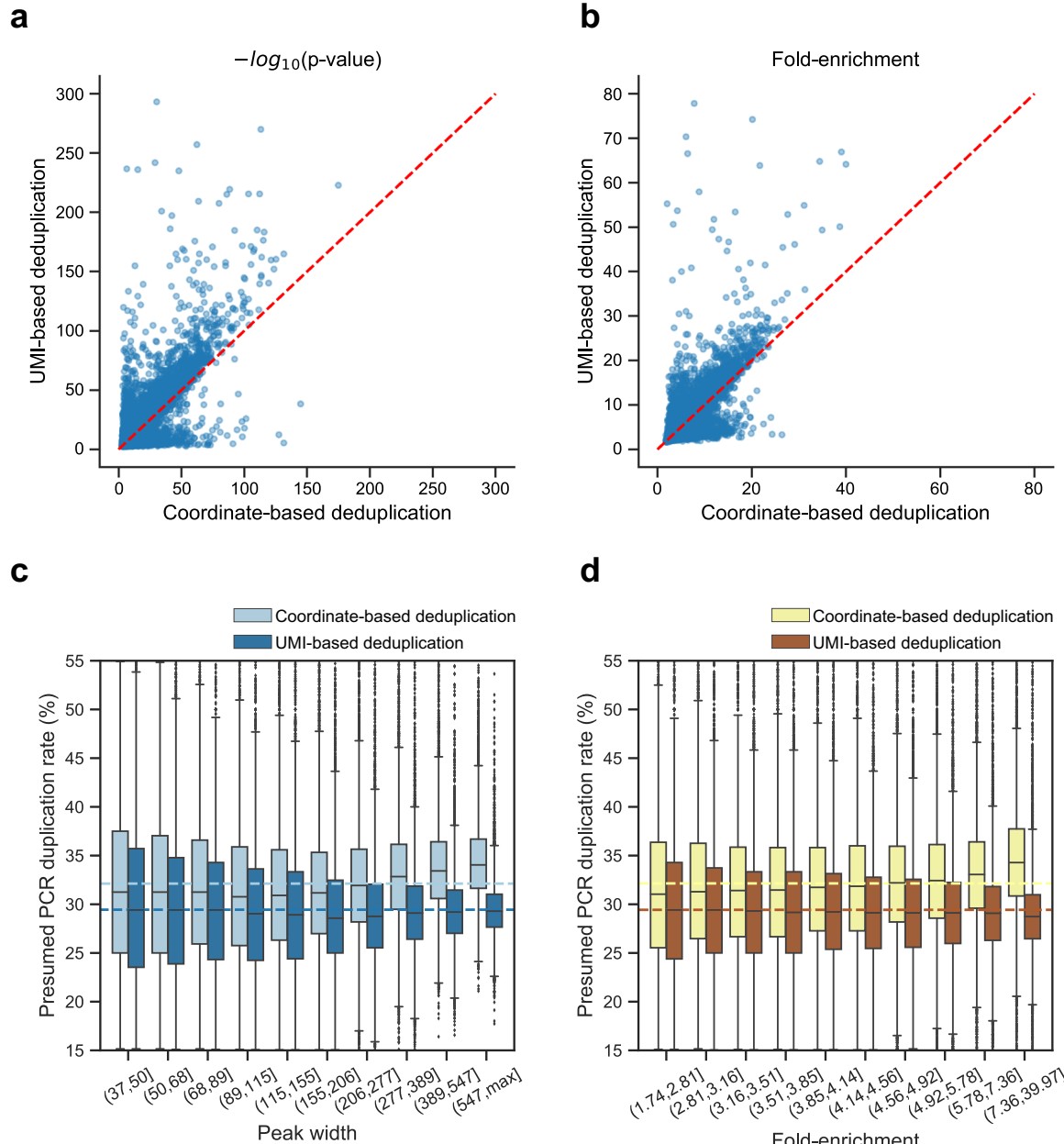

**Fig. 4 The impact of deduplication in different ways on peak calling.** All analyses were performed using UMI-ATAC-seq data from rice sample C019.
**a**, **b** Scatter plots comparing the *P*-values and fold-enrichment values of overlapping peaks (overlapping each other by at least 50%) in CD and UD datasets.
**c**, **d** Box plots illustrating the distribution of the presumed PCR duplication rates for peaks with different peak widths or fold-enrichment values. We divided the peak width or fold-enrichment into deciles for analysis. The dashed lines represent the averages of the corresponding presumed PCR duplication rates.

of the most accessible regions. Compared to the CD mode, we indeed observed better footprint patterning in the UD mode (Supplementary Fig. 6b). Taken together, these results suggest that UMI-ATAC-seq can significantly improve the sensitivity of footprint identification.

The UD mode can identify more footprints than the CD mode, but are the UD-only footprints biologically significant? We compared the UD-only footprints with potential transcription factor binding motifs identified by FIMO ($P$-value $< 10^{-5}$). The common and UD-only footprints overlapped with the FIMO motif sites by about 30 and 25%, respectively (Fig. 6b). To evaluate whether the overlap between the footprints and the FIMO results is random, we employed a shuffling-based approach. The results

show that both common and UD-only footprints overlap significantly with the FIMO motif sites compared to the random ($P$-value $= 1.1 \times 10^{-140}$ and $4.9 \times 10^{-34}$, respectively) (Fig. 6a), suggesting that many of the additional footprints identified by UMI-ATAC-seq are biologically meaningful.

## Discussion
ATAC-seq has proven to be an efficient method to analyze open chromatin and transcription factor footprints[26–28]. However, both published data and our data show an increase in the presumed PCR duplication rates in highly accessible chromatin regions, suggesting that the conventional way to identify PCR

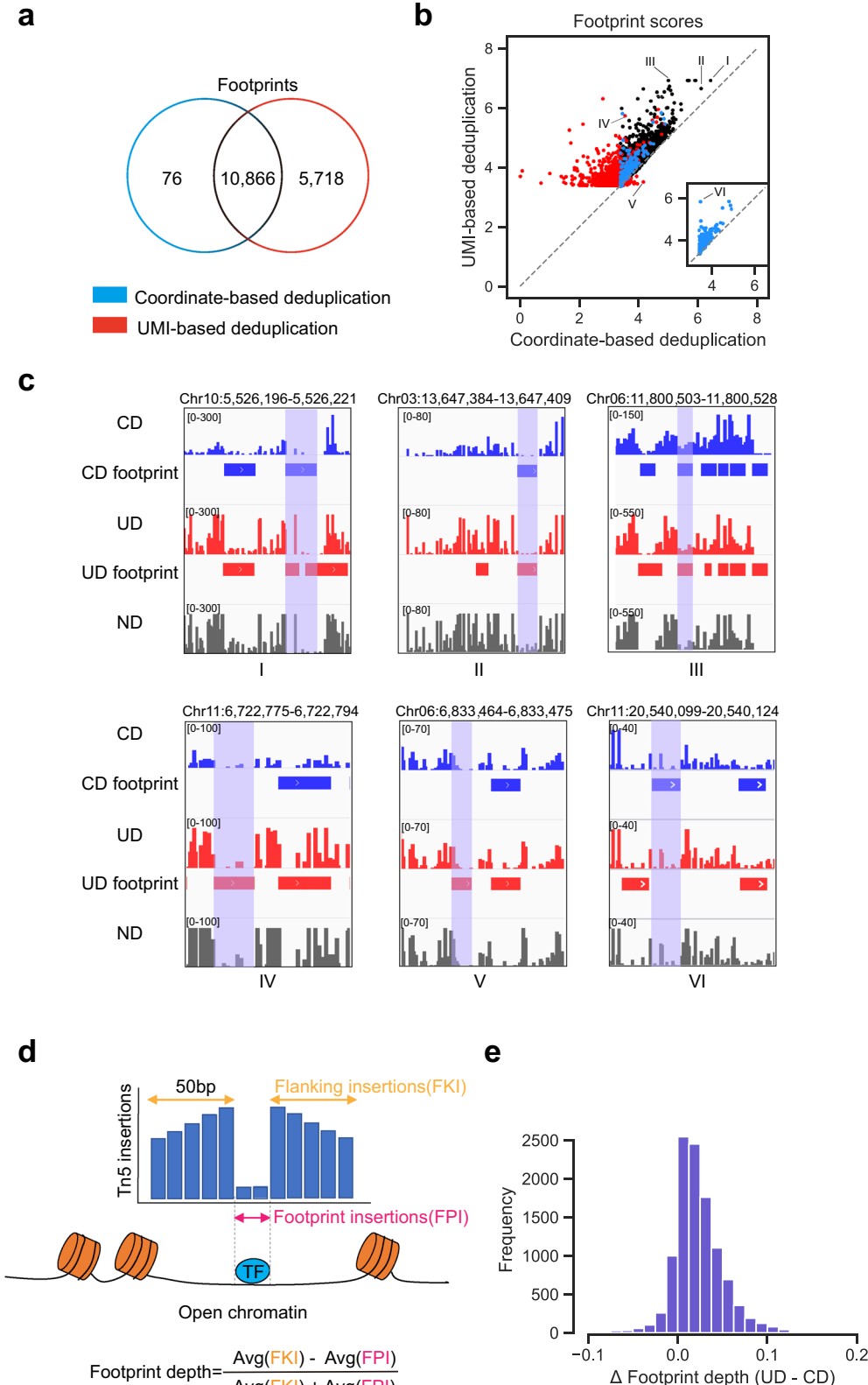

**Fig. 5 UMI-ATAC-seq improves the sensitivity of footprint identification. a** Venn diagram displaying the footprints identified by pyDNase (Wellington Footprint *P*-value < $10^{-30}$) with CD and UD datasets. The common footprints are defined as those overlapped by at least one base. **b** The relationship of footprint scores calculated based on CD and UD datasets. The footprint scores are calculated as log($-\log_{10}(P$-value)). The colors of points are the same as the Venn diagram in a. The inset shows the CD-only footprints. **c** The Tn5 insertion frequencies around some identified footprints in CD (blue), UD (red), or ND (gray) datasets. The genomic coordinates of the shaded areas are shown above each plot. **d** Schematic diagram illustrating the footprint depth (FPD) calculation. "Avg" represents the average function. **e** The distribution of the differences in FPD values between CD and UD datasets. Footprints identified in CD datasets were used.

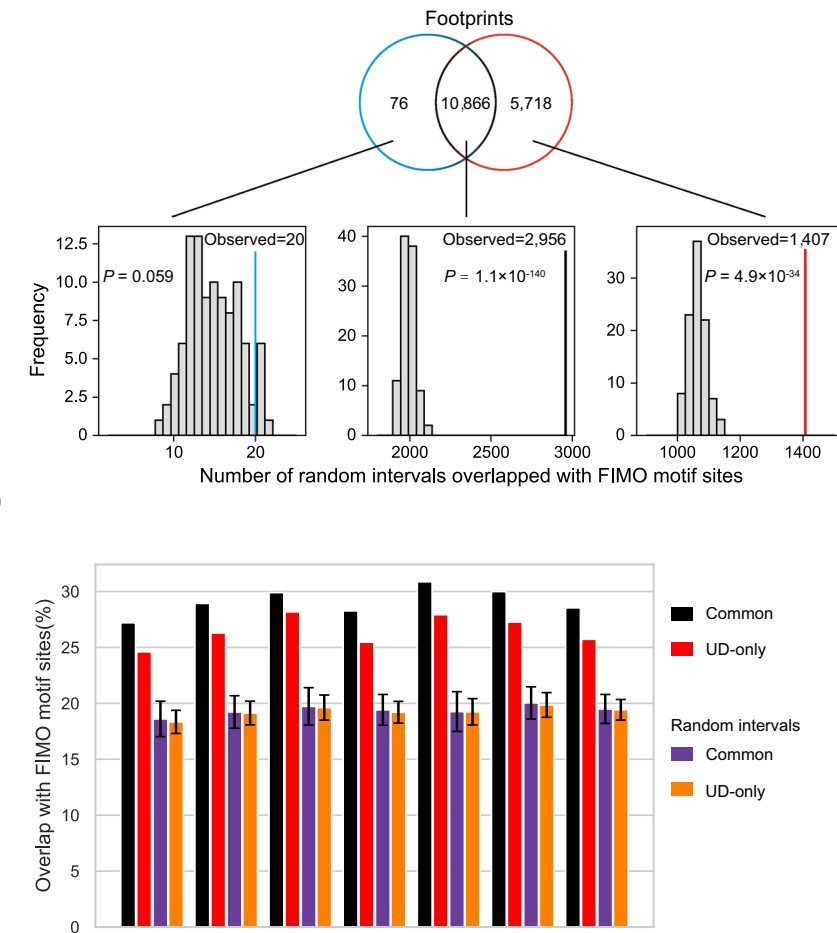

**Fig. 6 The overlap between identified footprints and FIMO motif sites.** The distribution of three types (CD-only, common, and UD-only) of shuffled intervals overlap (by at least 50%) with FIMO results. The vertical lines indicate the number of footprints identified by pyDNase overlapped with FIMO motif sites. **b** Bar plot illustrating the ratio of footprints or random intervals overlap with FIMO results in different samples. Error bars depict the 95% confidence interval around mean values at each shuffling-based test ($n = 100$).

duplicates has adverse effects on quantification and footprint identification in ATAC-seq. Footprint identification often requires deep sequencing of the libraries. It could be straightforward to infer that as the sequencing depth increases, the probability of generating independent but identical fragments would increase. Such "natural duplicates" would be erroneously removed as PCR duplicates in conventional ATAC-seq, diminishing the benefit of deep sequencing and flattening the footprint pattern. This might be one of the reasons why a previous study reported that the number of footprint would reach the limit at moderate library depths[15]. Our UMI-ATAC-seq technique has successfully adapted the standard ATAC-seq procedure to incorporate UMIs so that it could accurately distinguish between PCR and natural duplicates, which would be expected to improve the footprinting efficiency in deep sequencing.

In addition, many genomic assays construct sequencing libraries based on Tn5 transposase[17]. Thus, the idea of our method can theoretically be adapted to these relevant technologies, such as methyl-ATAC-seq[29], Trac-looping[30], and ATAC-see[31].

Finally, an inconvenience to our method might be the extraction and processing of UMI sequences. To simplify this process, we have developed and made available a relevant pipeline. We anticipate that UMI-ATAC-seq could enhance the potential of ATAC-seq applications.

## Methods

### UMI-ATAC-seq procedures

*Tn5 production.* Commercial Tn5 transposase is available from Illumina or many companies. Alternatively, in our study, in-house Tn5 was purified according to the procedure published by Picelli[17] with little modification. pTXB1 clones were transformed into BL21 cells (NEB) for production. Cells were grown in LB with ampicillin at 37 °C until OD600 = 0.6. The culture was chilled to 23 °C, and the expression of Tn5 was induced by adding 1 mM IPTG. The culture was then grown overnight at 23 °C and harvested by centrifugation. The cells were resuspended by HEGX (20 mM HEPES-KOH at pH 7.2, 0.8 M NaCl, 1 mM EDTA, 10% glycerol, 0.2% Triton X-100), with complete protease inhibitors (Roche) and lysed by sonication. 10% Polyethyleneimine was added to remove nuclei acid. The supernatant was loaded on a 10-mL chitin column (NEB). The column was washed with HEGX, and Tn5 was cleaved from the intein by adding 100 mM DTT and incubated for 36–48 h. The elution was tested using a Bradford assay. The fractions with the strongest blue color were pooled and dialyzed versus two changes of 1 liter of 2X Tn5 dialysis buffer (100 HEPES-KOH at pH 7.2, 0.2 M NaCl, 0.2 mM EDTA, 2 mM DTT, 0.2% Triton X-100, 20% glycerol).

*Tn5 transposon complex assembly.* All the oligonucleotides were synthesized in Shenggong Shanghai (Supplementary Table 1). Illumina sequencing systems utilize the first 25 cycles of Read 1 for cluster identification and quality metrics calculations. In order to adapt this requirement, we designed mixed adapters of four lengths to insure enough base diversity in the first 25 cycles. ME-A6/13/20/25 and ME-B were first annealed to MErev respectively to form double-stranded MEDS-As and MEDS-B. Then the four MEDS-A6/13/20/25 in different lengths were mixed with equal molar as a mix of MEDS-As. Then 1:1 ratio of MEDS-A mixture and MEDS-B are assembled with in-house Tn5 under room temperature for 1 hour to form the final transposon. The transposon complex can be stored at −20 °C.

*Preparation of rice panicles cell nuclei.* Fresh rice panicles were cut into small pieces and the nuclei were separated by chopping with 500 µL chopping buffer (15 mM Tris-HCl pH7.5, 20 mM NaCl, 80 mM KCl, 0.5 mM spermine, 5 mM 2-ME, 0.2% TritonX-100). After staining with DAPI, the sample was sorted by BD Aria SORP flow cytometry, and 100,000 nuclei were collected into a 1.5 mL centrifuge tube. The nuclei were centrifuged at $500 \times g$ under 4 °C and enriched by removing the supernatant, then stored on ice.

*Library construction and sequencing.* The library construction was adapted from human ATAC-seq protocol[4] with little modification. The tagmentation reaction was carried out by adding 2 µL Tn5 complex, 8 µL 5x reaction buffer, and 30 µL ddH$_2$O into the separated rice nuclei and incubated at 37 °C for 30 min. The transposed DNA fragments were purified by TaKaRa MiniBEST DNA Fragment Purification Kit (No. 9761). Adapters were added by PCR amplification using NEBNext® High-Fidelity 2X PCR Master Mix (M0541). Supplementary Table 1 listed the PCR primer sequences, and different indexes were incorporated by the 8 bp barcode sequences in Barcoded PCR Primer 2. After 10 cycles of PCR amplification, the libraries were pooled and purified by AMPure beads. Then the libraries were size selected at 180–600 bp using PippinHT (HR00187). The final library was sequenced by with TruSeq sequencing primer (Supplementary Table 1).

**UMI-ATAC-seq raw data processing and alignment.** Raw reads were first trimmed by Trimmomatic (v.0.36)[32], with parameters of a maximum of 2 seed mismatches, a palindrome clip threshold of 30, and a simple clip threshold of 10. Reads shorter than 30 bp were discarded. We then extracted the first 6 bp in each read pair as the UMI sequence and added it to the information line of the read pair. We next located and trimmed the sequence before the AGATGTGTATAAGA-GACAG (ME) sequence in Read 1. Trimmed reads were aligned to the *Oryza sativa L.ssp.japonica* (cv.Nipponbare) reference genome (v.7.0)[33] using BWA-mem with default settings, and the reads mapped to mitochondrial and chloroplast DNA were filtered. We only used properly paired reads with high mapping quality (MAPQ score > 30) for further analysis. Sorting aligned reads and removing PCR duplicates with mapping coordinates were conducted using SAMtools (v.1.9)[34]. We considered the reads with the same mapping coordinates but with different UMI sequences as rescued reads. The Tn5 insertion positions were determined as the start sites of reads adjusted by the rule of "forward strand +4 bp, negative strand −5 bp"[4].

**Processing Arabidopsis and human ATAC-seq data.** The ATAC-seq raw data in human (cell line HEK293) were downloaded from NCBI SRA (SRX3511086) and aligned to the human genome assembly GRCh38. The ATAC-seq data in Arabidopsis whole seedlings were downloaded from NCBI SRA (SRX2000808) and aligned to the Arabidopsis Genome (TAIR10). Tn5 insertion events were counted with no deduplication (ND) and coordinates-based deduplication (CD) separately. The PCR duplication rate was calculated as (ND − CD)/ND.

Peak calling were performed on Arabidopsis and human data using the "callpeak" function in MACS2[19] with the same following parameters: "--shift 100 --extsize 200 --nomodel --keep-dup all -B --SPMR --call-summits" except with different "-g" (1.87e9 and 2.7e9, respectively). Footprints were identified using "*wellington_footprints.py*" script in the pyDNase package[23] with ATAC-seq mode ("-A" parameter) and a threshold of *P*-value < 10$^{-30}$.

**Annotating the rescued reads.** The reads rescued by UMI were aligned to the *Oryza* Repeat Database[35] using blastn (v.2.5.0) with the following parameters: "-evalue 1e-10 -num_alignments 1".

**Peak calling.** Peak calling were performed on reads with coordinates-based deduplication (CD) and UMI-based deduplication (UD) separately, using the "callpeak" function in MACS2[19] with the following parameters: "-g 3.0e8 --nomodel --keep-dup all -B --SPMR --call-summits". To evaluate the impact of deduplication modes on peak calling, sets of "--shift" and "--extsize" parameters were tested. The specific parameters are marked in Supplementary Fig. 5. We used the parameters "--shift 15" and "--extsize 38" for the subsequent analysis.

**Identifying transcription factor footprints.** After calling peaks in CD and UD datasets separately, we merged the two sets of peaks and extended the peak intervals with the length <100 bp to 100 bp. The merged peaks were used to identify footprints by the "*wellington_footprints.py*" script in the pyDNase package[23]. Another script "*dnase_wig_tracks.py*" was employed to output the frequency of Tn5 insertion events, and IGV (Integrative Genomics Viewer)[24] was utilized for visualization. The scripts were used in ATAC-seq mode ("-A" parameter) with other parameters in default values. For footprints identified only in the CD or UD datasets, we merged the two sets of footprints and then calculated the footprint scores with both UD and CD datasets using the same intervals of footprints. The footprint scores were transformed from Wellington *P*-value[23] using the following

formulas:

$$P - \text{value} = F\left(\text{FP}^+, \text{FP}^+ + \text{SH}^+, \frac{l_\text{FP}}{l_\text{FP} + l_\text{SH}}\right) \times F\left(\text{FP}^-, \text{FP}^- + \text{SH}^-, \frac{l_\text{FP}}{l_\text{FP} + l_\text{SH}}\right)$$

$$\text{Footprint score} = \log(-\log_{10}(P - \text{value}))$$

where $l_\text{FP}$ is the length of a possible footprint (11–26 bp); $l_\text{SH}$ is the length of shoulder on each side of the footprint (fixed at 35 bp); FP$^+$ (forward strand) and FP$^-$ (negative strand) are the total number of Tn5 insertions inside the footprint. SH$^+$ (upstream shoulder in forward strand) and SH$^-$ (downstream shoulder in negative strand) are the total number of Tn5 insertions inside the shoulder of the footprint. $F$ is the binomial cumulative distribution function.

**Calculating footprint depth values.** We calculated footprint depth values with the following formula:

$$\text{Footprint depth} = \left(\frac{\text{FKI}}{100} - \frac{\text{FPI}}{L}\right) \Big/ \left(\frac{\text{FKI}}{100} + \frac{\text{FPI}}{L}\right)$$

where *FKI* is the total number of Tn5 insertions in the 50 bp upstream and the 50 bp downstream of the footprint; *FPI* is the total number of Tn5 insertions within the footprint, and *L* is the length of the footprint.

**Scanning potential binding sites of transcription factors.** We identified potential binding sites of transcription factors using FIMO[25] (v.5.0.5) with the default parameters except that the parameter "thresh" differed for different applications. The position weight matrices (PWMs) were downloaded from JAS-PAR[36] (http://jaspar.genereg.net/downloads/).

**Assessing the overlap between the identified footprints and FIMO motif sites.** A shuffling-based approach was used to evaluate whether the overlap between the identified footprints and FIMO motif sites is random. Briefly, we randomized the footprint intervals in peaks using bedtools shuffle[37]. Then we estimated the new overlapping rate with FIMO results and repeated the above steps 100 times to obtain a distribution. The distribution was used to perform Z-test for the observed overlapping rate.

**Statistics and reproducibility.** All statistics and analyses were performed using the software packages listed in Methods. Statistical Z-test were conducted in R using the *z.test* function in the TeachDemos package. Correlations between the CD, UD, and ND datasets are Pearson correlation coefficients. All UMI-ATAC-seq data can be downloaded from NCBI and the statistical results can be reproduced according to the parameters given in Methods.

**Reporting summary.** Further information on research design is available in the Nature Research Reporting Summary linked to this article.

## Data availability
UMI-ATAC-seq data from rice young panicles are available at the NCBI BioProject (PRJNA602139); ATAC-seq data from Arabidopsis thaliana and human (cell line HEK293) were obtained from the NCBI SRA Database (SRX2000808 and SRX3511086, respectively).

## Code availability
Code to remove sequencing adapters and PCR duplicates with UMIs can be obtained from the GitHub repository (https://github.com/tzhu-bio/UMI-ATAC-seq).

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

## Acknowledgements

We thank Mr. Jiacheng Li for advice on data analysis, and Mr. Qinghua Zhang for the support in sequencing. This work was supported by grants from The National Key Research and Development Program of China (2016YFD0100803), the National Natural Science Foundation of China (31922065, 31771755), and the Fundamental Research Funds for the Central Universities (2662016PY065).

## Author contributions

W.X. conceived and designed the project. R.Z., K.L., and C.X. performed experiments. T.Z. and K.L. analyzed sequencing data. T.Z. and W.X. wrote the paper with input from all other authors. All the authors reviewed and approved the paper.

## Competing interests

W.X. has filed a provisional patent application on UMI-ATAC-seq. All other authors declare no competing interests.
