## [Peer Review File · Communications Biology]

Reviewers' comments:

Reviewer #1 (Remarks to the Author):

In this short report, Zhu et al developed a new method to introduce unique molecular identifiers (UMIs) to ATAC-seq. Compared to standard ATAC-seq protocol, the new approach can distinguish independently generated identical Tn5 insertion events from PCR duplicates. Therefore, the new approach can rescue ~6% of the total mapped reads. Because of the rescued reads, the authors claimed that the footprint identification could be improved (~50% more footprints were identified). The paper is well written and the analysis is clearly demonstrated. I only have one question. For the CD-only footprints (blue dots in Fig. 1c), why are the footprint scores obtained from UD higher than from CD? If the scores from UD data are higher than CD, why the footprints were only called in the CD dataset, but not in the UD dataset? The authors should explain it. Is it possible that the calculation for footprint scores is inconsistent in UD and CD? Also, why do we need an inset in Fig. 1c? There is not an explanation in the Figure legend.

Reviewer #2 (Remarks to the Author):

ATAC-seq is a powerful technology to measure open chromatin sites, which contain important regulatory sites on the genome. As a sequencing technology, ATAC-seq also faces a problem of distinguishing and removing identical reads resulted from PCR amplification from those natural duplicates with the same sequence but coming from different fragment (so, different chromosomes or cells). This work presented a simple but effective method, UMI (or random barcoding), to accomplish this. This strategy, random barcoding, has been frequently practiced in many sequencing technologies, like RNA-seq, and sometimes in ChIP-seq and CLIP-seq. It is no surprise that it also works for ATAC-seq.

The following are some comments and questions:

1, as mentioned above, the novelty of using UMI in ATAC-seq library construction sounds limited. But interestingly, this strategy, as far as I know, is not commonly used in ATAC-seq studies. Why this is the case? So I think the main contribution of this paper could be a comprehensive evaluation to show how it improves ATAC-seq studies. For example, the authors should better incorporate detailed distribution of PCR duplication rate vs. sequencing depth and peak intensity. It could be a scatter plot with x-axis of sequencing depth and y-axis the peak intensity (or normalized by width). Every peak is a dot with colors representing the duplication rate. The sequencing depth can be simulated by down-sampling from a library with very deep sequencing.

2, It would be more convincing to show some statistics of how a good deduplicating affects peak calling of ATAC-seq data, as peaks are usually the biologically meaning output from an ATAC-seq library.

3, it would be even better to analyze what kind of peaks are more affected by deduplicating. I would imagine a narrow peak is more easily affected.

4, figure 1d needs a little explanation. What are the red and blue blocks under the insertion tracks? Are they footprints? Importantly, how can I tell the red tracks and blocks are better than blue ones? If the authors could prove the red tracks identifies more and correct TF footprints, that'll be very good.

Minors:

1. P2-48

The correlation between ND and UD is higher than CD and UD. There must be some factors effecting the correlation results. The total reads number can be a potential factor that caused difference. If the read number of UD is much larger than CD, then the correlation analysis between CD and UD is less significant. The followed parallel analysis between CD and UD would be suspected. As there is higher correlation between ND and UD and authors implied that large

quantity of footprints is the golden standard. Then I would query the necessity of deduplication for ND-CD group.

2. Fig1d

What are the bars below the ATAC-seq reads track? It seems that they are not genes or peaks. And there is not statement on them. Please add the legend.

The shadow also lacks the coordinated legend.

3. Fig1 c, d

It seems that ND group and CD group are much more similar in IGV tracks. Then the analysis in fig1c could be repeated for ND group. It may help trace the difference.

Reviewer #1:

In this short report, Zhu et al developed a new method to introduce unique molecular
identifiers (UMIs) to ATAC-seq. Compared to standard ATAC-seq protocol, the new approach
can distinguish independently generated identical Tn5 insertion events from PCR duplicates.
Therefore, the new approach can rescue ~6% of the total mapped reads. Because of the
rescued reads, the authors claimed that the footprint identification could be improved (~50%
more footprints were identified).

The paper is well written and the analysis is clearly demonstrated. I only have one question.

For the CD-only footprints (blue dots in Fig. 1c), why are the footprint scores obtained from
UD higher than from CD? If the scores from UD data are higher than CD, why the footprints
were only called in the CD dataset, but not in the UD dataset? The authors should explain it.
Is it possible that the calculation for footprint scores is inconsistent in UD and CD?

**RESPONSE 1.1: We appreciate the opportunity to further clarify this section of our**
**manuscript. We identified footprints using pyDNase with Wellington algorithm. The**
**calculation for footprint scores is consistent in UD and CD. They were calculated by the**
**formula as mentioned in Methods: “Identifying transcription factor footprints”. For the**
**CD-only footprints, the corresponding footprint scores in UD datasets were actually**
**higher than in CD datasets. The CD-only footprints could not be identified in UD**
**datasets merely because they are likely omitted by the Wellington algorithm, which**
**identified other close footprints around and therefore prevented the identification of the**
**relevant footprints. We have added additional text (page 9, lines 213-221) to address this**
**comment and explained this algorithm in details below:**

**The Wellington algorithm calculate p-values for every base pair in the peak intervals**
**based on the distribution of the Tn5 insertions, where the p-values are assigned to the**
**base pair at the center of the potential footprint. A potential footprint center has several**
**p-values, for the algorithm searching footprints with widths from 11 to 26 bp by default.**
**Finally, the Wellington algorithm picks up the footprint with the lowest p-value in a**
**region. Therefore, using either the CD or UD datasets, we might obtain different**
**footprints in the same region. For the CD-only footprints, although the footprint has the**

**lowest p-value in CD datasets for that region, other footprints in that region might be**
**identified with the lowest p-value in the UD datasets. As a result, the UD dataset would**
**identify other close footprints around CD-only footprints (e.g., region VI in Fig.1d). In**
**this case, if some reads are rescued with UMI, the scores of footprints obtained from UD**
**datasets could be higher than from CD datasets, even for the CD-only footprints.**

Also, why do we need an inset in Fig. 1c? There is not an explanation in the Figure legend.

**RESPONSE 1.2: The inset shows the CD-only footprints, for a better display of their**
**distribution. We have added this explanation in the Figure legend.**

Reviewer #2

ATAC-seq is a powerful technology to measure open chromatin sites, which contain
important regulatory sites on the genome. As a sequencing technology, ATAC-seq also faces a
problem of distinguishing and removing identical reads resulted from PCR amplification from
those natural duplicates with the same sequence but coming from different fragment (so,
different chromosomes or cells). This work presented a simple but effective method, UMI (or
random barcoding), to accomplish this. This strategy, random barcoding, has been frequently
practiced in many sequencing technologies, like RNA-seq, and sometimes in CHIP-seq and
CLIP-seq. It is no surprise that it also works for ATAC-seq.

1, as mentioned above, the novelty of using UMI in ATAC-seq library construction sounds
limited. But interestingly, this strategy, as far as I know, is not commonly used in ATAC-seq
studies. Why this is the case? So I think the main contribution of this paper could be a
comprehensive evaluation to show how it improves ATAC-seq studies. For example, the
authors should better incorporate detailed distribution of PCR duplication rate vs. sequencing
depth and peak intensity. It could be a scatter plot with x-axis of sequencing depth and y-axis
the peak intensity (or normalized by width). Every peak is a dot with colors representing the
duplication rate. The sequencing depth can be simulated by down-sampling from a library
with very deep sequencing.

**RESPONSE 2.1: Thanks for the comments. Although UMI is often used in**

high-throughput sequencing methods, UMI is not commonly used in ATAC-seq studies,
probably because its significance for ATAC-seq applications has not yet been realized. In
addition, the applicability of UMI in ATAC-seq needs to be carefully designed to make it
easy to use and allow for widespread applications. We realized the deduplication issue in
the standard ATAC-seq method and developed the UMI-ATAC-seq method, which is
easy to grasp. In our design, all steps are the same as for conventional ATAC-seq, except
that the PCR primers need to be replaced when constructing sequencing libraries. The
UMI-ATAC-seq library is compatible with standard sequencing on the Illumina
platforms and flexible enough to allow mixing with other kinds of samples. To simplify
the data analysis process, we have developed and made available an analytic pipeline.
For these advantages, we believe that UMI-ATAC-seq could enhance the potential of
ATAC-seq applications.

For the second comment, since PCR duplication rate, sequencing depth and peak
intensity varied from peak to peak, we performed similar analysis in samples but
avoiding the need of down-sampling. The results could be found in Figs. 1, 3 and 4,
showing that computational removal of PCR duplicates based only on their identical
mapping coordinates introduces increasing bias for quantifying highly accessible
chromatin regions, and UMI-based deduplication could avoid such bias.

2, It would be more convincing to show some statistics of how a good deduplicating affects
peak calling of ATAC-seq data, as peaks are usually the biologically meaning output from an
ATAC-seq library.

**RESPONSE 2.2:** We thank the reviewer for this comment and agree that this deserved
to be analyzed. We called peaks in CD and UD datasets separately and compared the
peak numbers, p-value and fold-enrichment. This analysis and its results are now
described on page 8, lines 180-197. The results are visualized in Fig. 4.

“To evaluate the impact of incorporating UMIs on peak calling, we called peaks using
MACS2 with different parameters reported by different studies. We compared the
number of peaks, p-values and fold-enrichments obtained with datasets processed in CD
and UD modes, respectively. And we found only small differences in the number of

peaks between the CD and UD datasets for various peak calling parameters
(Supplementary Fig.4).

However, for most of the highly significant peaks, the results of the UD dataset are
slightly better than the CD dataset in term of the p-value and the fold-enrichment
(Fig.4a, b). Furthermore, we found that wider peaks and peaks with higher
96 fold-enrichment had more presumed PCR duplicates in CD mode, and we could
eliminate such bias in UD mode (Fig.4c, d). These results suggest that while
UMI-ATAC-seq has only a slight effect on peak calling, it might improve the estimation
of fold-enrichment of peaks, as it can eliminate PCR deduplication bias.”

3, it would be even better to analyze what kind of peaks are more affected by deduplicating. I
would imagine a narrow peak is more easily affected.

**RESPONSE 2.3:** We have added relevant results to Fig. 4, see **RESPONSE 2.2** as well.

4, figure 1d needs a little explanation. What are the red and blue blocks under the insertion
tracks? Are they footprints?

**RESPONSE 2.4:** Thanks for pointing this out. We have added relevant labels to the
Figure legend (now Fig. 5c).

Importantly, how can I tell the red tracks and blocks are better than blue ones? If the authors
could prove the red tracks identifies more and correct TF footprints, that'll be very good.

**RESPONSE 2.5:** We appreciate the comment. We calculated the footprint depth (FPD)
values for ND, CD and UD datasets respectively and compared the differences of FPD
between each other. A bigger FPD value could be interpreted as decreased Tn5
insertions at the footprint region compared to its flanking regions. The results show that
FPD values in UD datasets are generally higher than both ND and CD datasets,
indicating that patterns of footprints are more apparent in UD mode. Moreover, to
evaluate whether the UD-only footprints are biologically significant, we compared the
UD-only footprints with potential transcription factor binding motifs identified by
FIMO with known position weight matrices. We found UD-only footprints overlapped

with the FIMO motif sites significantly. We have added this analysis and results to pages
9-10, lines 223-252:

“Moreover, we calculated the footprint depth (FPD) values for ND, CD and UD datasets
respectively and compared the differences of FPD between each other. A bigger FPD
value could be interpreted as decreased Tn5 insertions in the footprint region compared
to its flanking regions.(Fig.5d) The results show that FPD values in UD datasets are
generally higher than CD datasets, indicating that patterns of footprints are more
apparent in UD mode (Fig.5e, Supplementary Fig.6a). In addition, we scanned all
potential transcription factor binding motifs across the genome using FIMO with known
position weight matrices ($P\text{-value} < 1.0^{-4}$). We averaged the Tn5 insertions across all
predicted binding sites of transcription factor AP2/ERF and TCP1 that overlapped with
the common pyDNase footprints identified in both CD and UD modes and located in the
top 25% of the most accessible regions. Compared to the CD mode, we indeed observed
better footprint patterning in the UD mode (Supplementary Fig.6b). Taken together,
these results suggest that UMI-ATAC-seq can significantly improve the sensitivity of
footprint identification.

The UD mode can identify more footprints than the CD mode, but are the UD-only
footprints biologically significant? We compared the UD-only footprints with potential
transcription factor binding motifs identified by FIMO ($P\text{-value} < 1.0^{-5}$). The common
and UD-only footprints overlapped with the FIMO motif sites by about 30% and 25%,
respectively (Fig.6b). To evaluate whether the overlap between the footprints and the
FIMO results is random, we employed a shuffling-based approach. The results show
that both common and UD-only footprints overlap significantly with the FIMO motif
sites compared to the random ($P\text{-value}=5.14 \times 10^{-139}$ and 2.03×10^{-33} , respectively, Fig.6a),
suggesting that many of the additional footprints identified by UMI-ATAC-seq are
biologically meaningful.”

Minors:

1. P2-48

The correlation between ND and UD is higher than CD and UD. There must be some factors
effecting the correlation results. The total reads number can be a potential factor that caused
difference. If the read number of UD is much larger than CD, then the correlation analysis
between CD and UD is less significant. The followed parallel analysis between CD and UD
would be suspected. As there is higher correlation between ND and UD and authors implied
that large quantity of footprints is the golden standard. Then I would query the necessity of
deduplication for ND-CD group.

**RESPONSE 2.6: We confirmed that the correlation between ND and UD is higher than**
**CD and UD in all samples. The explanation is that the conventional coordinate-based**
**deduplication approach introduces severe bias to the highly accessible regions, which**
**have greater weight in the calculation of Pearson's correlation coefficient. Our results**
**suggest that UD is the best choice, compared with CD and ND. Please see RESPONSE**
**2.5 for details.**

2. Fig1d

What are the bars below the ATAC-seq reads track? It seems that they are not genes or peaks.

And there is not statement on them. Please add the legend.

The shadow also lacks the coordinated legend.

**RESPONSE 2.7: Thanks for pointing this out. We have modified Fig. 5c and added**
**relevant explanation to the legend.**

3. Fig1 c, d

It seems that ND group and CD group are much more similar in IGV tracks. Then the analysis
in fig1c could be repeated for ND group. It may help trace the difference.

**RESPONSE 2.8: We apologize for the confusion. We now labeled IGV tracks (Fig. 5c) to**
**make it clear. Regarding the comparisons between CD, ND and UD, please see**
**RESPONSE 2.5.**

REVIEWERS' COMMENTS:

Reviewer #1 (Remarks to the Author):

The authors addressed my concerns and the paper is ready for publication.

Reviewer #2 (Remarks to the Author):

The revision is convincing and thorough. I have no more concerns.

Reviewer #1:

The authors addressed my concerns and the paper is ready for publication.

**RESPONSE: We thank the reviewer for the comments.**

Reviewer #2:

The revision is convincing and thorough. I have no more concerns.

**RESPONSE: We thank the reviewer for the comments.**
